# Stereotypes and Prejudices in Nursing Prison Activities: A Reflection

**DOI:** 10.3390/healthcare11091288

**Published:** 2023-04-30

**Authors:** Guido Vittorio Travaini, Francesco De Micco, Francesca Biscella, Elisa Carminati, Emma Flutti, Francesca Garavaglia, Laura Marino, Arianna Zini, Roberto Scendoni, Anna De Benedictis

**Affiliations:** 1School of Medicine, Vita-Salute San Raffaele University, 20132 Milan, Italy; 2Research Unit of Bioethics and Medical Humanities, Department of Medicine and Surgery, Università Campus Bio-Medico di Roma, Via Alvaro del Portillo, 21, 00128 Roma, Italy; 3Department of Clinical Affair, Fondazione Policlinico Universitario Campus Bio-Medico, Via Alvaro del Portillo, 200, 00128 Roma, Italy; 4Department of Law, University of Macerata, 62100 Macerata, Italy; 5Research Unit of Nursing Science, Department of Medicine and Surgery, Università Campus Bio-Medico di Roma, Via Alvaro del Portillo, 21, 00128, Roma, Italy

**Keywords:** ethics of care, quality and safety of care, prison, nursing, prejudices

## Abstract

Background: In the prison environment, the nursing profession has particularly complex peculiarities and aspects, so much so that prison nurses require advanced specialist skills and specific education. Can nurses’ stereotypes and prejudices in prison settings affect nursing care? What are nurses’ perceptions of the prison environment and people in detention? This study aims, on one hand, to outline the figure of the nurse in the prison environment and current regulations and, on the other hand, to explore whether and how stereotypes and prejudices may affect the way care is provided. Methods: Starting with an analysis of the literature, the authors administered a questionnaire to a group of nurses who shared data and reflections. Results: This study sheds a new light on nursing in the prison environment, exploring how nurses’ stereotypes and prejudices may affect the care of patients. Conclusions: It would be desirable to develop research in this field to enable a more conscious approach to a world that is still considered distant and dangerous, and to overcome the misperceptions and prejudices that may negatively affect the way of caring.

## 1. Introduction

The nursing profession frequently encounters critical situations, often very different from each other, which require behavioral choices, most often decisive for the user. Critical situations can be identified as ethical dilemmas, which imply decision making. However, by whom are these decisions made? Like all health professionals, today certainly more than yesterday, nurses are called upon to choose the best possible course for the patient and to respond with competence, relevance, responsibility, and timeliness [1].

To safeguard this, the Code of Ethics for Nurses states that the nurse is active in the analysis of ethical dilemmas experienced in daily practice, even in the event of conflicts arising from different ethical views because respecting fundamental human rights and the ethical principles of the profession is an essential condition for its exercise [2]. The Code constitutes the guiding elements for the ethical practice of nurses, emphasizing the importance of ethical reflection in the humanization of care [3].

In the prison environment, the nursing profession has particularly complex peculiarities and aspects, so much so that the prison nurse requires advanced specialist skills and specific education [4]. Working in a prison means, above all, adapting to the environment and its rules, which significantly influence the organization of work itself. 

This is the starting point for the following research questions that guided the present article: Can nurses’ stereotypes and prejudices in prison settings affect nursing care? What are nurses’ perceptions of the prison environment and people in detention?

The study aims, on one hand, to outline the figure of the nurse in the prison environment and the current regulations and, on the other hand, to explore whether and how stereotypes and prejudices may affect the way care is provided.

### 1.1. Background

An instrument that helps nurses to establish and define the rules of conduct that must necessarily be observed in the exercise of specific professional nursing activities is the Code of Ethics. It contains the ethical principles of the profession and explains how nurses must act in accordance with fundamental human rights. However, the Code of Ethics does not always identify and provide a solution to all ethical dilemmas and, above all, it does not consider the variables that arise in certain places, such as prisons. The correctional nurse provides care in a locked facility to incarcerated individuals and may be faced with difficult challenges and decisions. In prison, the difficulties of nursing care are due not only to the possible limitations caused by the many prejudices and stereotypes today towards prisoners and those who decide to work with them, but also to the ethical aspects. Firstly, “the role of correctional nursing may not be understood, recognized or appreciated by the nursing profession in general. Working with incarcerated populations and providing care to people who have committed crimes can make nurses uncomfortable. Correctional nurses report that, as a specialty, they felt isolated from other specialties due to lack of recognition and stigmatization” [5].

### 1.2. The Constitutional Arrangement of Prison Health

The “state of health” does not only affect the individual because it also reflects on the community. In this way, its protection “implies and includes the duty not to harm or endanger the health of others by one’s own behavior”. The right to health is configured as a supreme constitutional value, traceable to the person’s mental and physical integrity. This meaning is the result of a gradual jurisprudential evolution, as also inferred from the construct of Articles 2, 3, 27 paragraph 3 and 32 of the Italian Constitution.

The social mandate of prison healthcare includes two obligations to balance, which are control, i.e., healthcare is limited to responding to individual requests for medical services, and care, i.e., healthcare promotes a culture of health, performs a preventive function, including institutional hygiene, and ensures adequate care of the person, considering the greater difficulty in accessing care and often greater vulnerability in custodial settings [6]. Consequently, health professionals are required to provide quality care by respecting constitutionally recognized rights and expanding access to means and services for prevention, care, and health recovery [7]. 

### 1.3. Ethical Dilemmas of Nurses in Places of Detention: Prejudices and Stereotypes

As much as the code of ethics and bioethics give guidelines on how to behave, nurses encounter ethical dilemmas. This happens when values conflict regarding what is the best mode of patient care. Due to this, nurses may also experience tension and frustration in their daily clinical practice and professional dissatisfaction. Nurses also report a low level of job satisfaction and a lack of training and specific skills [8]. Likewise, personal and patient relationships may be compromised. An additional critical issue is the time frame within which such decisions must be made. Prison nurses, compared to community and hospital nurses, face additional barriers because of the social stigma associated with the correctional healthcare system, constantly finding themselves exposed to criticism from the community itself. According to a recent qualitative study conducted in Northern Italy on a sample including 31 correctional nurses [4], it is clear how the collision between prison security and nursing care invalidates the opportunity to provide adequate care for prisoners. Specifically, the authors found five themes that seem to clash with prison nursing practice: prisoners’ healthcare needs; negotiation between custody and care; satisfaction of working in prisons; obstacles to quality care; and safety, with which the theme of manipulation adopted by prisoners is transversally associated. In this perspective, manipulation can lead to further moral dilemmas and ethical issues, arising from the sense of frustration experienced in a yet compromised nurse–patient therapeutic relationship. Prison nurses are expected to care for every inmate equally and without prejudice, regardless of their crime, but this is not always possible. Specifically, they are requested to struggle between custody and caring, being nonjudgmental, and remembering the importance of boundaries [9]. Clinical ethics do not provide standardized solutions, but they do offer a method for the professionals to learn how to process their own analysis and reach a reasoned conclusion facing a problematic situation, according to the dimensions of the ethics of a job well done [10]. Over the past few years, several methods have been developed to overcome this problem, which includes, for instance, the comparison with clinical cases made by bioethicist Sandro Spinsanti [11] and the four-question model developed by Fry and Johnston (2004) [1].

Prejudice is defined as an idea or opinion conceived on personal beliefs and general thoughts, without direct knowledge of facts, people, and things, such that it strongly influences evaluation and thus leads to error [12]. Regarding the prison environment, inmates are exposed to socially originated hardships and negative phenomena, such as prejudice and stereotypes. Prisoners are defined by almost all of society as scum, those to be kept away because they are dangerous for multiple reasons. In addition, the prison environment and forced cohabitation can be an amplifying cause of marginalization and discrimination even among cellmates themselves. Stereotypes are false, frequently negative representations of reality. They change the perception of the external world by stiffening judgments, which remain unchanged over time and are difficult to modify and criticize. Stereotypes find application in all spheres of human life; nevertheless, they are widely present in correctional settings. First of all, in the detention of foreigners, there is the recurring idea that “as the number of foreigners in Italy increases, so does the number of criminals”. In addition, we find other common stereotypes, such as the sentiment “if a person kills, it is not right for them to be in prison for a short time, but they should pay dearly”. This scenario suits the label theory developed by sociologist Howard Becker [13], according to which society is the most important factor in the identification of deviance (it is the community itself that creates deviance). Stereotypes, prejudices, and preconceptions can be corrected with the right knowledge, (re)education, awareness, and humanization. Adopting and promoting a human rights or social justice approach to correctional healthcare has the potential to improve understanding of the social determinants of crime and health [14]. Based on the Code of Ethics (especially Article 4), healthcare practitioners should be free of all stereotypes and prejudices: “the nurse provides care to the person beyond everything”. It is in this contest that strong contradictions arise. On one hand, the profession morally imposes a lack of prejudice; on the other hand, the professional as a human being who also carries personal baggage of preconceived ideas about the subject in front of them, and especially about the crime committed by them. A study by Crampton (2014) [15] showed that knowing the prisoner’s crime affected the ability to meet nursing goals. Nurses are not likely to know the crime their patient committed; however, this information could be readily accessible to them through the internet [16]. A question arises in this regard: is it right for a doctor and/or nurse to know what crime that inmate has committed? The answer is subjective; however, in case of a positive answer, the fallout on the method of care and the attitude held toward the patient would not be negligible.

### 1.4. Ethics and Bioethics

In healthcare, ethics is the standard value system used in the interpretation and analysis of each clinical situation. The values by which the healthcare profession is inspired are those of health as a fundamental human right to be safeguarded and protected. Reference is made in the Code of Ethics to the “conscience clause”. The nurse, therefore, acts according to the principles of beneficence, autonomy, justice, truthfulness, fidelity, and confidentiality. As for bioethics, it assumes relevance in the context of the principles and values expressed in the Code of Ethics itself. It is identified as an interdisciplinary subject that covers a wide area, considering it can represent a limit to what people can do to prevent their self-destruction and/or the destruction of the environment in which they live. The issue of the right to health for prisoners acquires special ethical significance if one considers health as a complete physical, mental, and social well-being [17] because of the higher vulnerability to which they are exposed and their precarious state of health (prior, contemporaneous, and subsequent to detention). Prison healthcare professionals and correctional officers do not have the right nor the obligation to make decisions on prisoners’ health. Community healthcare professionals may need to highlight the need of prisons’ patients to receive ethically appropriate care just like non-incarcerated patients [18,19]. However, the principle of equal opportunity between prisoners and free people to access the good of health on one hand encounters obstacles in the demands of security; on the other hand, it comes into contradiction with the practice of detention which produces suffering and illness itself. The National Bioethics Committee has already considers the subject, highlighting the dramatic conditions of overcrowding in Italian prisons and the worrying issue of suicide within those contexts. Despite this, the National Bioethics Committee has still not clearly mentioned the problem of the well-known irreconcilable incompatibility of the prison system with the right to health, which is becoming more and more evident day by day [20].

### 1.5. Research Objectives

The aim of this study was to explore the types of prejudices, stereotypes, and ethical dilemmas that nurses may experience regarding the care of people held in prison and to explore whether and how these prejudices, may affect the way of caring for patients. 

## 2. Materials and Methods

### 2.1. Study Design

To answer the research questions, a multicenter survey was conducted using a structured questionnaire developed and administered to a sample of nurses from Italian hospitals.

### 2.2. Participants

The participants in the study were nurses from various hospitals in Lombardy. The criterion for inclusion in the study was the possession of membership of the Professional Order of Nurses, although it was not necessary to have previous work experience in the prison environment. There was no selection among the healthcare professionals; each of them had free remote access to the compilation of the administered questionnaire.

### 2.3. Data Collection

In relation to the specific objective of the study, a structured questionnaire, consisting of 17 multiple-choice questions, was drawn up (Table 1). 

The questionnaire was reviewed for face and content validity by four experts from a panel of experts in the sector, including a forensic pathologist, a penitentiary nurse, and a psychologist, starting with the results of the literature review. Panel members were asked to evaluate each statement for clarity, ease of use, and appropriateness [21].

The first part concerned biographical data in which personal reference data were collected (i.e., gender, age, reference region, and marital status). The second part concerned professional training, i.e., educational qualification, period of practice, and previous work experience as a prison nurse (time, place, and experience). Finally, the third part focused on ethical dilemmas, prejudices, and stereotypes.

The questionnaire was developed and administered via Google Forms via e-mail to 600 nurses. Three monthly reminders were sent via e-mail.

Each questionnaire contained a brief introduction, explaining the objectives of the study and how the data would be collected. The questionnaires were completed during the month of October in 2021. Data collection was carried out by means of remote compilation by the participants. This mode was adopted due to both the restrictions of the COVID-19 pandemic, ensuring complete confidentiality for participants, and convenience in administering the questionnaire.

### 2.4. Ethical Considerations

The study was conducted in accordance with the principles of the Declaration of Helsinki developed in Brazil [22]. Data were collected in an aggregate manner with the utmost anonymity and in full compliance with Italian Privacy Law [23]. 

## 3. Results

A total of 78 questionnaires were obtained with a response rate of 13%. Once the data was collected, the questionnaires were viewed one by one, and the data obtained were manually entered into an Excel file and analyzed using descriptive statistics methods.

The final sample included 78 nurses (response rate 13%), of which 85.9% were female. The sample characteristics are summarized in Table 2, and Table 3 reports the responses to the structured questions.

Most of the respondents were aged between 18 and 55 years, and 87.2% of them had been nursing for more than five years. Only four nurses had previous work experience as a health worker in the prison health service, while the remaining 94.9% of respondents stated that they had never carried out such work; three of them had worked for less than a year in an adult prison and the other had for more than five years. Two of them considered working in a penitentiary tiring from a psychological point of view; one of them considered it an experience that everyone should have. One nurse rated the experience as useful for the profession.

Considering those who have not experienced work in a detention institution, 51.3% of them had never taken it into consideration, 44.7% considered it to be a useful and formative experience, while three nurses (3.9%) did not consider it useful nor formative. According to 53.8% of the sample, the type of offence was a factor that could influence the mode of caring. According to 11.5%, age was a factor, 2.6% believed ethnicity was a factor, 2.6% believed the extent of the sentence was a factor, 1.3% believed the gender of the prisoner was factor. For the other 28.2%, none of the items indicated in the questionnaire influenced the mode of caring.

The crimes that, more than others, were considered to affect the mode of caring were pedophilia (66.7%), homicide (14.1%), family abuse/violence (11.5%), and crimes of criminal association (7.7%). With the same treatment, 80.8% of the sample would treat everyone first, regardless of the offence committed. A. total of 12.8% would firstly treat a person detained for crime association, 3.8% would be firstly treat a person detained for a crime of homicide, and 2.6% would firstly treat a person detained for crimes of family abuse. If a prisoner admits their guilt to a crime, for the 69.2%, the mode of caring would not change; meanwhile, for 25.6%, the mode of caring would change according to the type of offence committed and for 5.1%, the mode of caring would change regardless. A total of 29.5% did not believe that working in prison is more dangerous in terms of personal risk, while 29.5% indicated that this depends on the type of detention center, 26.9% thought it is more dangerous regardless, and 14.1% indicated that it depends on the type of offenders.

Finally, in regard to the Code of Professional Standards, 65.4% of the sample believed that this does not totally help in resolving ethical dilemmas, while 20.5% thought that it totally helps and 14.1% indicated that it does not help at all.

## 4. Discussion

The daily practice of healthcare is fraught with perplexity and obliges nurses and doctors to make choices and decisions in which important moral values come into play [24]. The values that inspire the nursing profession are those of health, understood as a fundamental human right to be safe-guarded and protected. 

However, studies addressing nurses’ experiences of moral problems in daily practice indicate that pressure to work within time limits and manage a heavy workload adds to burnout and ethical insensitivity, thus compromising real-world perspective in nursing encounters.

From the data collected from the questionnaire, it emerged that those who had work experience in prison found it useful and regarded it as an experience everyone should have; meanwhile, most of the applicants who had not experienced working in a prison had never considered it in their lives. This is reflected in the many jobs offered in prisons for nurses that were not taken into consideration. The third section of the questionnaire on ethical dilemmas, prejudices, and stereotypes showed that a large proportion of the participants would be influenced in terms of the type of care provided by the type of offence committed by the prisoner and that only 28% of the candidate nurses would be influenced neither by the type of offence nor by the size of the sentence, age, gender, and even by ethnicity.

In support of the results obtained from the questionnaire, several studies conducted on nursing care provided in correctional settings have shown that personal and public judgments about people who are incarcerated and who deserve healthcare can influence nurses’ decision making and practice. This is also because some nurses in prisons are aware of the legal charges against their patients. This can make it extremely difficult to provide objective care, because such knowledge can create a biased perspective and diminish the quality of care provided. Therefore, although it is difficult to ignore, the extent of the crime or alleged crime must be considered irrelevant by a correctional nurse [14].

A question that may arise spontaneously from the results of the questionnaire is whether it is right for a doctor and/or nurse to know what crime a prisoner has committed. The answer is subjective, although certainly as both people and health professionals it can have a huge influence on nurses’ perception of others knowing the crime they have committed; thus, this may also influence methods of care and attitudes towards them in some way. It would be interesting to investigate if and how stereotypes, prejudices, and preconceptions can be corrected with the right knowledge, (re)education, awareness raising, and humanization. When asked whether the Code of Nursing Ethics helps in the resolution of ethical dilemmas, it emerged that 65.4% of the candidates thought that the Code of Nursing Ethics did not help at all in the resolution of ethical dilemmas. On the basis of these results, it would be appropriate, in order to complete the study, to investigate the specific motivation of the answers. However, Article 3 of the Nurses’ Code of Ethics mentions: “the nurse provides care of the assisted person, with respect for dignity, freedom, equality, his life choices and conception of health and well-being, without any social, gender, sexual orientation, ethnic, religious, and cultural distinction. The nurse abstains from any form of discrimination and blame towards all those he meets in his work” [2]. 

This is very often not easy for the professional because they are in close contact with human discomfort, pain, suffering, and guilt. The patient/prisoner is a patient category that encompasses multiple dynamics, characteristics, and problems which, when applied to the prison context, can make the nurses’ work difficult. In fact, although the prison is a territorial context in which the nurse can provide care, very often no training is provided for this environment, which requires health interventions relative to the demand of health needs [25,26].

## 5. Conclusions

The present study shows that only a limited number of subjects had previous experience in the prison environment and that many of the subjects had never considered it. This result highlights how the prison environment is still without direct knowledge and is often full of prejudices that lead healthcare personnel to disregard it.

The data also show that most nurses would be influenced by the type of crime committed by the patient, especially in the case of pedophilia, and this could have an impact on caring behavior. This is very important because it shows how, even though the Code of Ethics teaches nurses to take care of everyone without distinction, in particular environments such as prisons, personal ideas and moral codes unfortunately take over, which can create inner contrasts.

In addition to this, positive aspects also emerged, for example, most of the nurses involved in the study would treat all patients regardless of the crime committed, and only the remaining 20% would be affected.

The starting point, in fact, is that preparation for the nursing profession devotes little attention to this sphere, which determines in health workers poor real-world knowledge of the problems and dynamics present in it. This lack of knowledge is also confirmed by the present study. Many subjects in the sample surveyed stated that they considered prison to be a dangerous environment. This statement shows that preconception prevails over reality. In fact, more episodes of violence can occur in hospital emergency rooms than in prisons, as the latter are places subject to stricter forms of control. Additionally, nurses stressed the need for specific education that would help them manage ethical issues. Indeed, specific training emphasizes the special nature of correctional nursing [4]. In addition to the need to provide ethical training for prison workers, the identification of the real needs underlying the healthcare of the prison population and the promotion of collaboration between prison care and the healthcare system seem to be pre-eminent [27]. On this matter, a study conducted in the USA by Oyolu (2022) [28] opened a starting point for future discussions. From this research, it emerges that, in the USA, correctional nursing has not received much attention from nursing programs. This causes negative perceptions linked to the correctional environment which, consequently, lead to a decrease in commitment to and enthusiasm in being a nurse working in correctional settings. To face this issue, Oyolu (2022) [28] suggested initiating clinical visits for nursing students in a detention center to reduce prejudices. Including prison healthcare as a care environment in student modules could deflate bias and negative perceptions.

The study presents some limitations mainly related to the limited number of nurses involved and the use of a non-validated and structured questionnaire. Future research might consider a larger sample in order to perform construct validity of the realized tool, as well as the use of interviews and semi-structured questionnaires to allow for a more in-depth exploration of the phenomenon.

### Relevance to Clinical Practice

This study highlights how, to date, there is still a lack of literature on the nursing profession in the prison environment. Therefore, it would be very desirable and useful to have more information and experience in this field in the future, which would enable a more conscious approach to an environment that is still considered distant and dangerous. It is crucial to foster the transition from a simple and cursory perception to deeper knowledge and greater awareness of what the prison environment is so as to overcome, in part (or even better, completely), the misperceptions, preconceptions, and prejudices that risk undermining or distorting a correct way of caring.

## Figures and Tables

**Table 1 healthcare-11-01288-t001:** Questionnaire outline.

Items and Related Survey Questions	
1. Please indicate your gender	Female
	Male
	I prefer not to specify
2. Please indicate your age	18–35 years
	36–55 years
	56–70 years
3. Please indicate your region of residence	*Multiple choice of all Italian regions*
4. Please indicate your educational qualification	Primary school certificate
	Secondary school certificate
	High school diploma
	Vocational school diploma
	Bachelor’s degree
	Master’s degree or single-cycle degree
	Master’s degree and/or PhD and/or specialization diploma
	Other qualification not indicated or foreign qualification
	No qualification
5. Marital status	Single
	Married
	Separated
	Divorced
	Widowed
6. How long have you been working as a nurse?	Less than 1 year
	From 1 to 5 years
	More than 5 years
7. Have you ever had previous experience working as a healthcare professional in a prison environment?	Yes
	Never
8. If yes, for how long?	Less than 1 year
	From 1 to 5 years
	More than 5 years
9. If yes, where?	Adult correctional institution
	Juvenile correctional institution
	Other
10. If you have never experienced it, do you find an experience in a detention institution useful and educational?	Yes
	No
	I have never considered it
11. Which of the following could influence your caring behavior?	Type of offence
	Extent of punishment
	Age
	Gender
	Ethnicity
	None of these
12. Which of these types of crime do you think may affect your caring behavior?	Pedophilia
	Homicide
	Family abuse
	Drug trafficking
	Criminal association offences (mafia)
13. Given the same need for treatment, whom would you treat first?	A person held for a murder offence
	A person convicted of pedophilia
	A person convicted of a crime of family abuse
	A person convicted of an association offence
	Everyone regardless of the offence committed
14. Do you think our code of ethics helps in resolving these ethical dilemmas?	Yes
	No
	Not totally
15. If a prisoner confided to you that he was guilty of a crime, would you change your caring behavior?	Yes, in any case
	No, in any case
	Depends on the type of offence
16. Do you think working in prison is more dangerous in terms of personal risk?	I do not think so
	It depends on the type of facility
	Depends on the type of prisoner
	Yes, regardless
17. Other observations or personal opinions with respect to the topic under study	*Free text response*

**Table 2 healthcare-11-01288-t002:** Sample characteristics.

Characteristics	*n.*	*%*
Female	67	85.9
Male	11	14.1
Age		
18–35 years	22	28.2
36–55 years	51	65.4
56–70 years	5	6.4
Region		
Lombardy	73	93.6
Piedmont	2	2.6
Tuscany	2	2.6
Liguria	1	1.3
Education		
High school diploma	4	5.1
Vocational school diploma	26	33.3
Bachelor’s degree	34	43.6
Master’s degree or single-cycle degree	2	2.6
Master’s degree and/or PhD and/or specialization diploma	6	7.7
Other qualification not indicated or foreign qualification	6	7.7
Marital status		
Single	29	37.2
Married	42	53.8
Separated	2	2.6
Divorced	4	5.1
Widowed	1	1.3
Years of experience as a nurse		
From 1 to 5 years	10	12.8
More than 5 years	68	87.2

**Table 3 healthcare-11-01288-t003:** Main results.

Items and Related Survey Questions		*n.*	*%*
Have you ever had previous experience working as a healthcare professional in a prison environment?	Yes	4	5.1
	Never	74	94.9
If yes, for how long? *[only referred to the 4 nurses who had previous experience in a prison environment]*	Less than 1 year	3	75
	From 1 to 5 years	0	0
	More than 5 years	1	25
If yes, where? *[only referred to the 4 nurses who had previous experience in a prison environment]*	Adult correctional institution	3	75
	Juvenile correctional institution	0	0
	Other	1	25
If you have never experienced it, do you find an experience in a detention institution useful and educational? *[only referred to the 74 nurses who did not have previous experience in a prison environment]*	Yes	33	44.7
	No	3	3.9
	I have never considered it	38	51.3
Which of the following could influence your caring behavior?	Type of offence	42	53.8
	Extent of punishment	2	2.6
	Age	9	11.5
	Gender	1	1.3
	Ethnicity	2	2.6
	None of these	22	28.2
Which of these types of crime do you think may affect your caring behavior?	Pedophilia	52	66.7
	Homicide	11	14.1
	Family abuse	9	11.5
	Drug trafficking	0	0
	Criminal association offences (mafia)	6	7.7
Given the same need for treatment, whom would you treat first?	A person held for a murder offence	3	3.8
	A person convicted of pedophilia	0	0
	A person convicted of a crime of family abuse	2	2.6
	A person convicted of an association offence	10	12.8
	Everyone regardless of the offence committed	63	80.8
Do you think our code of ethics helps in resolving these ethical dilemmas?	Yes	16	20.5
	No	11	14.1
	Not totally	51	65.4
If a prisoner confided to you that he was guilty of a crime, would you change your caring behavior?	Yes, in any case	4	5.1
	No, in any case	54	69.2
	Depends on the type of offence	20	25.6
Do you think working in prison is more dangerous in terms of personal risk?	I do not think so	23	29.5
	It depends on the type of facility	23	29.5
	Depends on the type of prisoner	11	14.1
	Yes, regardless	21	26.9
Other observations or personal opinions with respect to the topic under study	*Free text response*		

## Data Availability

The data presented in this study are available on request from the corresponding author.

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
