# Peer review of "Stereotypes and Prejudices in Nursing Prison Activities: A Reflection"

_healthcare, 2023, doi:10.3390/healthcare11091288_

Round 1

Reviewer 1 Report

Please see document attached

Author Response

Responses to Editor’s and Reviewer’s comments to “The role of prison nursing between care and ethical dilemmas”. 

Dear Editor and Reviewers,

We are very grateful for the positive response received concerning our manuscript and the possibility to review and resubmit it. We confirm our strong commitment to contribute to Healthcare. Following your indications, we addressed the comments in a new version of the manuscript.

We hope to have addressed the revisions appropriately and that our manuscript has improved significantly. All changes made are marked in red.

Please find detailed replies (marked in red) to Reviewers comments about the changes made.

Response to Reviewer #1

Thank you for giving me the opportunity to review the article ‘The Role of Prison Nursing between Care and Ethical Dilemmas‘. The aim of this paper was to identify the ethical dilemmas that underlie nursing practice in a penitentiary setting. The authors set out to determine if personal opinions and prejudices might influence the way in which nursing care is delivered. Prejudices and preconceptions were highlighted as possible reasons that might affect the care provided by nurses to prisoners. The authors rightfully suggested that more research is needed in this area. I found the topic interesting and novel.

Thank you for your kind words.

General Comments

I was very keen to review this article as I felt the topic of this paper might attract a wide readership due to its novelty.

Thank you for your kind words.

However, I found the manuscript confusing to read, starting with the title which could be better defined. Also, it is suggested that the title better reflect the type of article so that the reader gets an indication of what to expect from the manuscript.

Thanks for this comment. The title has been revised to better reflect the type of article.

This was still not clear from the abstract, which should be more concise.

Thanks for this comment. The abstract has been revised.

I would comment that the article was not well structured. It appears that the authors attempted to fit too much in one article. There seems to be some component of a narrative literature review and then another with primary data collection. It is suggested that the authors choose one type of article to focus on, and then do this in a better structure. As it currently reads this work appears as a thesis ‘squashed’ into a single paper.

Thanks for this comment. The structure of the article has been completely revised and simplified.

There were several sentences that had to be read and reread in order to comprehend its meaning. Thus, there seems to be a formatting or English language problem. This manuscript needs to be clearer and presented in a well-structured manner.  It is clear that a lot of work has gone into this research, and with major reworking, there may be a good publication from it.

Thanks for this comment. The English language has been completely revised.

Methods and methodology

There is room for improvement in the experimental design to make it more appropriate to test the hypothesis in this research.

Thanks for this comment. There is room for improvement in the experimental design to make it more appropriate to test the hypothesis in this research.

If a systematic review was done then the structure of that type of research should be presented. This was not the case in this work. For example, there was no indication of article numbers PRISMA diagrams, quality assessments checks, inclusion and exclusion criteria etc. The completeness of the review topic covered needs improvement.

Thanks for this comment.  The paper has been revised and simplified. The reference to a systematic review of the literature, which was not carried out, has been deleted.

Participants (although de-identified) were used for part of this study. Therefore, the reviewer wonders why the ‘informed consent statement’ was ‘not applicable’?

Thanks for this comment.  A specific informed consent statement was not necessary, as a questionnaire was administered to a group of nurses exploring only experiences and perceptions related to their work. No patients were included in the study, and no areas directly related to patients were explored. As indicated in the paper, the study was conducted in accordance with the principles of the Declaration of Helsinki, and data were collected in an aggregate manner, in the utmost anonymity and in full compliance with the Italian Privacy Law.

There seems to be inconsistence about the validation of the questionnaire used.

Line 233: “The questionnaire was reviewed for face and content validity by four experts”.

Line 405: “the use of a non-validated questionnaire.”

Thanks for the comment. It has been better specified that the questionnaire created has not yet been validated with a construct validity and confirmatory factor analysis, but it has been assessed for content and face validity by a panel of experts.

There could be improvement in the data interpretation to make it appropriate and consistent throughout the manuscript.

Thanks for the comment. The paper has been revised.

Line 263: Is it possible that the response rate was 87% as opposed to 13%?

No, the response rate is correct: 13%.

Some of the explanations of the findings do not support what was indicated in the questionnaire. For example, Line: 276 “one of them considered it an experience that everyone should have……” Was there also a semi-structured questionnaire? The reviewer feels that this type of questionnaire would have been more suited for the objectives of this research. Here participants would have been able to provide more in-depth answers to open ended questions.

Thanks for this comment. The questionnaire was structured but there was a final open-ended question for any additional comments. Thanks for the suggestion, which has been included in the limits of the study and in future proposals.

Reviewer 2 Report

Introduction 

The introduction seems to me to be very coherent and complete. It has started by defining the study problem and the impact. It orders in a logical way the whole sequence of ideas about the object of study.

Objective: as part of the study is a literature review, perhaps some of the verbs used in the objectives should be to know, to describe, to explore. 

Method: this section needs improvement. 

Design: the study time, inclusion criteria for the literature review, type of research question and analysis for dealing with the bibliographies should be specified

Participants: it would be convenient to include exclusion criteria and if there were any problems with data collection. It would be convenient to know the ethical dilemmas experienced by nurses working in a penitentiary center if the sample had previous experience in this field. Or, if not, modify the objective of the study to one more in line with the results that could be obtained from the experience of working in a police custody environment. 

Data collection: the criteria for finalizing the collection and the size of the data needed to validate the questionnaire are missing. 

Data analysis: it is not specified how the quantitative data were analyzed or the algorithm used to analyze the literature review

Ethical considerations: I would add in this section the approval of the study by an ethics and research committee. 

Results 

The results need to be better developed; perhaps a table ordering the responses would be convenient to know the frequency and percentage of each variable in the study. I do not know if with these results it is possible to have a perception that is a more qualitative term of the care that nurses can provide in this area. It would be advisable for the authors to review the objectives of the study. 

Discussion 

In the discussion of the study, the authors express that from the results of the study it is clear that the work is considered an experience that everyone should have but that it is psychologically tiring, I do not find any question of the questionnaire that can emerge that data, only if it is useful or not useful. It would be convenient to extend this section of the study with a larger number of studies with which to discuss the research results.

Conclusion: I do not find in the results of the study the conclusion that nurses emphasize that they need specific training. This section has no citations and also has citations in two different styles. The authors may have mixed part of the discussion with the conclusions. 

references: there are references in different styles, check.

Author Response

Responses to Editor’s and Reviewer’s comments to “The role of prison nursing between care and ethical dilemmas”.

Dear Editor and Reviewers,

We are very grateful for the positive response received concerning our manuscript and the possibility to review and resubmit it. We confirm our strong commitment to contribute to Healthcare. Following your indications, we addressed the comments in a new version of the manuscript.

We hope to have addressed the revisions appropriately and that our manuscript has improved significantly. All changes made are marked in red.

Please find below detailed replies (marked in red) to Reviewers comments about the changes made.

Response to Reviewer #2

Introduction

The introduction seems to me to be very coherent and complete. It has started by defining the study problem and the impact. It orders in a logical way the whole sequence of ideas about the object of study.

Thank you for this comment.

Objective: as part of the study is a literature review, perhaps some of the verbs used in the objectives should be to know, to describe, to explore.

Thanks for this comment.  The paper has been revised and simplified. The reference to a systematic review of the literature has been deleted.

Method: this section needs improvement.

Thanks for the comment. The section has been revised.

Design: the study time, inclusion criteria for the literature review, type of research question and analysis for dealing with the bibliographies should be specified.

Thanks for this comment.  The paper has been revised and simplified. The reference to a systematic review of the literature has been deleted.

Participants: it would be convenient to include exclusion criteria and if there were any problems with data collection.

Thanks for this comment.  No exclusion criteria were envisioned. The participants in the study were nurses from various hospital centers in Lombardy. The criterion for inclusion in the study was the possession of membership of the Professional Order of Nurses.

It would be convenient to know the ethical dilemmas experienced by nurses working in a penitentiary center if the sample had previous experience in this field. Or, if not, modify the objective of the study to one more in line with the results that could be obtained from the experience of working in a police custody environment.

Thanks for this comment. The article was also revised with reference to the objectives of the study.

Data collection: the criteria for finalizing the collection and the size of the data needed to validate the questionnaire are missing.

Thanks for this comment. The small number of answers compared to the number of questions asked did not allow a validation study of the questionnaire through a construct validity. The questionnaire was evaluated through a content and face validity by a panel of experts. The research will also be extended in order to allow a future validation of the tool.

Data analysis: it is not specified how the quantitative data were analyzed or the algorithm used to analyze the literature review.

Thanks for this comment.  The paper has been revised and simplified. The reference to a systematic review of the literature has been deleted.

Ethical considerations: I would add in this section the approval of the study by an ethics and research committee.

Thanks for this comment. A specific approval for the manuscript by an Ethics Committee or Institutional Review Board was not necessary, as a questionnaire was administered to a group of nurses exploring only experiences and perceptions related to their work.

No patients were included in the study, and no areas directly related to patients were explored.

As indicated in the paper, the study was conducted in accordance with the principles of the Declaration of Helsinki, and data were collected in an aggregate manner, in the utmost anonymity and in full compliance with the Italian Privacy Law.

Results

The results need to be better developed; perhaps a table ordering the responses would be convenient to know the frequency and percentage of each variable in the study. I do not know if with these results it is possible to have a perception that is a more qualitative term of the care that nurses can provide in this area. It would be advisable for the authors to review the objectives of the study.

Thanks for this comment. The tables have been revised as suggested. The objectives of the study have been reformulated.

Discussion

In the discussion of the study, the authors express that from the results of the study it is clear that the work is considered an experience that everyone should have but that it is psychologically tiring, I do not find any question of the questionnaire that can emerge that data, only if it is useful or not useful. It would be convenient to extend this section of the study with a larger number of studies with which to discuss the research results.

Thanks for this comment. The discussion has been revised and improved.

Conclusion: I do not find in the results of the study the conclusion that nurses emphasize that they need specific training.

This section has no citations and also has citations in two different styles. The authors may have mixed part of the discussion with the conclusions.

Thanks for the comment. The conclusions have been revised and improved.

References: there are references in different styles, check.

Thanks for the comment. The references have been checked.

Round 2

Reviewer 1 Report

I thank the authors for the effort they have invested to significantly revise this article. They have adequately addressed the concerns I previously raised.

Reviewer 2 Report

Thank you for your contributions and modifications. I agree to the acceptance of the study